# COVID-19 in schools: Mitigating classroom clusters in the context of variable transmission

**Paul Tupper** [1]*, **Caroline Colijn** [1,2]

**1** Department of Mathematics, Simon Fraser University, Burnaby, Canada, **2** Department of Mathematics, Imperial College London, London, United Kingdom

* pft3@sfu.ca

**Data Availability Statement:** All code and data files are available in a GitHub repository: https://github.com/carolinecolijn/unfortunate-covid-events-schools.

**Funding:** PT was supported by a Natural Science and Engineering Research Council (Canada)

## Abstract

Widespread school closures occurred during the COVID-19 pandemic. Because closures are costly and damaging, many jurisdictions have since reopened schools with control measures in place. Early evidence indicated that schools were low risk and children were unlikely to be very infectious, but it is becoming clear that children and youth can acquire and transmit COVID-19 in school settings and that transmission clusters and outbreaks can be large. We describe the contrasting literature on school transmission, and argue that the apparent discrepancy can be reconciled by heterogeneity, or "overdispersion" in transmission, with many exposures yielding little to no risk of onward transmission, but some unfortunate exposures causing sizeable onward transmission. In addition, respiratory viral loads are as high in children and youth as in adults, pre- and asymptomatic transmission occur, and the possibility of aerosol transmission has been established. We use a stochastic individual-based model to find the implications of these combined observations for cluster sizes and control measures. We consider both individual and environment/activity contributions to the transmission rate, as both are known to contribute to variability in transmission. We find that even small heterogeneities in these contributions result in highly variable transmission cluster sizes in the classroom setting, with clusters ranging from 1 to 20 individuals in a class of 25. None of the mitigation protocols we modeled, initiated by a positive test in a symptomatic individual, are able to prevent large transmission clusters unless the transmission rate is low (in which case large clusters do not occur in any case). Among the measures we modeled, only rapid universal monitoring (for example by regular, onsite, pooled testing) accomplished this prevention. We suggest approaches and the rationale for mitigating these larger clusters, even if they are expected to be rare.

## Author summary

During the COVID-19 pandemic many jurisdictions closed schools in order to limit transmission of SARS-CoV-2. Because school closures are costly and damaging to students, schools were later reopened despite the risk of contact among students contributing

Discovery Grant, RGPIN-2019-06911 https://www.nserc-crsng.gc.ca/index_eng.asp They did not play any role in the study design, data collection and analysis, decision to publish, or preparation of the manuscript. CC was supported by a Genome BC grant, COV-142. https://www.genomebc.ca/ They did not play any role in the study design, data collection and analysis, decision to publish, or preparation of the manuscript.

**Competing interests:** The authors have declared that no competing interests exist.

to increased prevalence of the virus. Early data showed schools being mostly a low risk setting, but occasionally large outbreaks were observed. We argue that this heterogenous behaviour can be explained by variability in the rate of transmission, both at the level of the individual student and at the level of the classroom. We created a mathematical model of transmission in the classroom to explore the consequences of this variability for cluster size and control measures, considering what happens when a single infectious individual attends a classroom of susceptible students. We used the model to study different interventions with the aim of reducing the size of infection clusters, in situations where such clusters would be large. We found that interventions based on acting after symptomatic students receive a positive test, as is standard practice in many jurisdictions, are ineffective at preventing most infections, and instead found that only frequent screening of the entire class was able to reduce the size of clusters substantially.

## Introduction

Coronavirus disease 2019 (COVID-19) is a global pandemic caused by SARS-CoV-2, a newly emerged respiratory virus. While COVID-19 can be severe especially among the elderly, its impact on children and youth is relatively mild, with a very low fatality rate among children aged 0–19 years [1] and low levels of hospitalization and severe illness compared to adults. Children also comprise a lower portion of reported cases than they do of the general population in many settings, though they can get COVID-19 and can (at low rates) suffer complications [2]. To control the pandemic, many jurisdictions implemented widespread distancing measures including school closures, and partly as a result, despite the pandemic's global reach with over 40M cases worldwide [3] at the time of writing, there remains considerable uncertainty about the role of children and of schools in the transmission of COVID-19. A range of approaches to mitigating transmission in schools, in the context of circulating COVID-19 in the broader community, are being deployed as schools reopen for the fall and winter in many areas.

School closures prevent transmission by preventing contact among students and teachers; they also prevent onward transmission to family and friends. But the social costs of closures are too great for closures to be a feasible long-term strategy. School closures take a severe toll, often in ways that exacerbate inequality, disproportionately affecting children from marginalized groups [4]. Digital poverty and parents' limited availability to help pose difficulties with distance learning [5]. School closures impact physical and mental health as schools play a role in healthy eating, with school closures heightening food insecurity [6]. Disruption of social relationships and extracurricular activities impact mental health and time spent online may increase cyber-bullying [5, 7, 8]. Schools are a safe haven and safety net for children at risk of, or experiencing, domestic abuse, and the impacts of these are severe in both short and long terms [9]. So it is crucial to understand how much and by what route transmission occurs in the classroom environment, both so we can (i) accurately weigh the costs of school opening in terms of COVID-19 transmission against the immense social costs of school closure, (ii) determine which interventions are most effective in limiting transmission in the classroom environment.

A key area of uncertainty is just how transmissible COVID-19 is by students in the classroom environment. There are a number of studies following contacts of infected individuals known to have attended schools. Frequently, no or few new infections are observed, though there are exceptions. In Ireland in March 2020, there were no confirmed transmissions among

contacts of 3 students and 3 staff who were believed to be infectious; contacts included 924 children and 101 adults [10]. In Australia, contact tracing of 1448 contacts of 12 children and 15 adults with COVID-19 in 10 early childhood education settings found just 18 secondary cases. However, there was one early childhood setting outbreak in which an infected adult is thought to have infected 6 additional adults and 6–7 children leading to an overall attack rate of 35.1% [11]. In France, Fontanet et al. describe a cluster of COVID-19 in a high school setting in which the overall infection attack rate was 40.9% (in a retrospective serological study) [12], but in another study in the same setting, 3 students with COVID-19 had attended three different schools and there were no secondary cases; the authors concluded that there was no clear evidence of transmission in schools [13]. The European Centre for Disease Prevention and Control reviewed COVID-19 in children and schools [4] (Aug. 6, 2020); several of the 15 reporting countries surveyed reported clusters in educational settings; these were limited in size and were considered "exceptional" events. A review up to 11 May 2020 concluded that children are not likely to be primary drivers of the COVID-19 pandemic, but that they likely could transmit the virus [14]. These limited data detailing transmission in schools, together with the severe impacts that school closures have, especially in light of the exacerbation of existing inequality, make a good case for reopening schools with measures in place to minimize transmission.

However, the reason there have been so few transmission events may be more related to the lack of opportunity, rather than transmission being especially unlikely in school environments or among children. In most settings described above, community COVID-19 transmission was low and the spring-summer 2020 school session was short; there was little opportunity for exposure and transmission in schools. In many jurisdictions, strong measures were put in place that limited exposures in schools by reducing community prevalence and contact with school settings. For example, Sweden initially kept high schools open only on a distance basis, as did many Canadian provinces in June 2020; Denmark's elementary schools opened with distanced and smaller classrooms; South Korea shut down whole schools upon one student testing positive [15, 16]. In British Columbia, Canada, high schools remained closed and elementary school students optionally attended for 1–2 days per week. There were often only 5–6 students in classrooms intended for 20–30; summer reopening lasted for 3.5 weeks at a time when there were only on average 2–4 new reported COVID-19 cases per million population per day.

In contrast to the literature cited above, there have been reports of larger outbreaks and broader transmission among school-age children, particularly in jurisdictions with more community transmission at the time. In Georgia, USA, an overnight camp attended by 597 residents had a large outbreak resulting in 76% positivity among the 344 tested attendees [17]. The overall attack rate was 50% among those age 6–10 years and 44% overall, despite efforts to follow most components of the Centre for Disease Control's risk reduction recommendations (though masks were not worn by campers and there was "vigorous singing and cheering"). 26% of 153 cases for whom there was symptom data reported not having had symptoms. In Israel, all school classes reopened on 17 May 2020, with hygiene, face masks, health checks and distancing measures in place. A high school registered two unlinked cases ten days later; subsequently, school-wide testing found 153 students and 25 staff members who tested positive for COVID-19 with an additional 87 relatives and friends ultimately infected as well [18]. Cases were most concentrated in 7–9th grades with 17–30% of those year groups infected), compared to 1.6–4.5% among 10th-12th grade students. Nearly half of the 7–9th grade cases were asymptomatic. Classes were crowded and due to a heat wave, air conditioning was used and students were exempted from wearing masks. The age distribution of COVID-19 cases in Jerusalem also shifted, reflecting a higher portion of 10–19 year olds [18]. In Trois Rivières,

Québec, despite physical distancing and other measures, 9 of 11 students in an elementary class contracted COVID-19 after a student was infected in the community [19]. Also in Québec, 27 confirmed cases were found at a day camp [20], and subsequently caused nearly 20 secondary cases among mainly siblings, family and friends [21]. An outbreak in a Chilean high school began shortly after the first case in Chile was detected (March 3, 2020) [22]; by March 13 there were two confirmed cases in the school, which was then placed in quarantine. By April 6, 52 school community members had been confirmed positive and there had been one death. Serology in early May found antibody positivity rates of 9.9% among students and 16.6% among staff (compared to much lower baseline rates in the community) [22]. As of Sept. 25, 2020, 26 COVID-19 cases were linked to an outbreak in a Canadian elementary school [23]. A survey of school outbreaks in Germany [24] found 48 of them between the 28th of January and the 31st of August, comprising only (0.5%) of all the outbreaks in the country in that period. There were a total of 216 cases involved, almost half of which were adults (21 years or older). Most of the outbreaks (38/48) had 5 or fewer cases, though the largest had 25. There was no data available on the number of exposures without transmission, as an outbreak was defined by there being more than one case detected.

At the population level, children have been infected with COVID-19, though at lower rates than adults. In the EU/EEA and UK (as of 26 July 2020), only 4% of reported cases were among those under 18 (who comprise approximately 16% of the population) [4, 25]. In a large Spanish study, seroprevalence in children was under 3.1% compared to 6.2% among adults [26]. In a large Icelandic study of those at high risk, 6.7% of children under 10 and 13.7% of those over 10 tested positive [27]. A Swiss population-level study found only 1 of 123 children aged 5–9 year who tested positive for antibodies, suggesting a lower rate than other age groups, and did not find a lower rate in those aged 10–19 years than in the general population [28]. In the United States (where children comprise approximately 24% of the population overall), as of October 15, 2020, 10.9% of overall COVID-19 cases were in children (typically 0–18 or 19 years) with high variability by state [2]. 12 states reported > 15% of cases among children aged 0–19 year (0–20 in Tennessee) and two states reported under 5% of cases among children (typically 0–17 years in those states); in the United States schools have been closed for most of the pandemic thus far [2]. In Sweden, where elementary schools remained open, a study in May of 1100 individuals found that 4.4% of children and teenagers and 6.7% of adults aged 20–64 had antibodies; the relatively higher rate in children may suggest transmission in schools [29]. An analysis of over 575,000 contacts of 86,000 index cases in two states in India found enhanced transmission in similar-aged pairs, an effect that was strongest for those aged 0–14 and over 65 [30].

The two main public health concerns with respect to the transmission COVID-19 among children in schools are i) are we endangering children, teachers, staff and families by having children together in the classroom setting? and ii) does the presence of children in the classroom accelerate the spread of the virus through the broader community? The evidence above leaves the situation unclear. Large outbreaks are possible and children can transmit COVID-19 to each other [30], suggesting sufficiently high transmission rates for clusters to arise. On the other hand, there are many documented cases where there is exposure with little or no transmission, suggesting that frequently transmission is very low. This apparent inconsistency matches what we know about other COVID-19 transmission data: at one choir practice an individual infected 52/60 participants [31]; at another there appears to have been no transmission [10]. Occasionally multiple infections occur on a single flight [32]; but the majority of potential exposures on planes lead to no transmission [33]. These considerations point to COVID-19 transmission being highly heterogeneous, or "overdispersed", a phenomenon with a building body of evidence [30, 34–36].

Here we use stochastic individual-based simulations to explore the implications of the above observations for control of transmission clusters in classrooms. We consider two sources of transmission heterogeneity: individual variation in infectiousness and variability in how effective a particular environment/activity combination is for transmitting COVID-19. We include the potential for pre- and a-symptomatic transmission and for transmission outside of an identified set of close contacts (via aerosols and/or mixing outside of the group). We explore intervention protocols in the context of this heterogeneity, comparing interventions focusing on groups of close contacts to those intervening at the whole class level, and to those using wider regular testing.

## Methods

### Data

We use crowdsourced data available through Covid Écoles Québec [37] to inform our underlying simulation framework. They collect reports of known COVID-19 exposures or clusters in educational settings, along with the date, a date of last update, and the number of reported cases. Cases were only detected through a PCR test after the appearance of symptoms, so the reported clusters are likely underestimated in size, and many exposures and smaller clusters will be missed altogether. In Fig 1 we show the distribution of cluster sizes along with the type of school the cluster occurred in. The majority of exposures have led to no additional reported cases, which is indicated by clusters of size 1 in this data set. However, there is a tail of larger clusters. This data is consistent with a model of transmission where infectiousness is variable and the distribution of secondary cases is overdispersed. We model two contributing factors that are known to affect transmission [36]: the individual and the classroom/activity combination. Individuals vary extensively in viral load both over their course of infection and from individual to individual. In addition, talking, singing, shouting activities in crowded conditions in poor ventilation are associated with large reported outbreaks and with data on aerosol and droplet generation. We therefore model index cases of varying infectiousness arriving in classrooms whose additional contribution to transmission is variable, stratifying the simulations according to the individual and environment risks.

### Disease model

We model COVID-19 progression in an individual as having the states susceptible (S), exposed (E), presymptomatic (P), symptomatic (Sym), and recovered (R). Individuals start in the susceptible state and then transition to the exposed state when they are infected. Exposed individuals are not able to infect others. Individuals transition from the exposed state to the presymptomatic state at which point they are able to infect others, but have no symptoms. Individuals may either transition from presymptomatic to symptomatic states (showing symptoms while remaining infectious) or directly to the recovered state without ever showing symptoms. Symptomatic people eventually enter the recovered state, where they are no longer infectious. Presymptomatic or symptomatic individuals infect susceptibles at constant rate $\beta$ when they are together, where $\beta$ may depend on the individuals involved, their exact state, their proximity and the environment.

Individuals stay in the exposed state for the duration of the latent period, after which they become presymptomatic. Latent periods are modeled as gamma-distributed with mean $\mu_\ell$ and standard deviation $\sigma_\ell$. For each presymptomatic individual a gamma-distributed presymptomatic infectious period (PIP) and a gamma-distributed infectious periods are generated (with means and standard deviations $(\mu_P, \sigma_P)$ and $(\mu_i, \sigma_i)$ respectively). The individual is infectious for the duration of the infectious period, starting from when they enter the presymptomatic

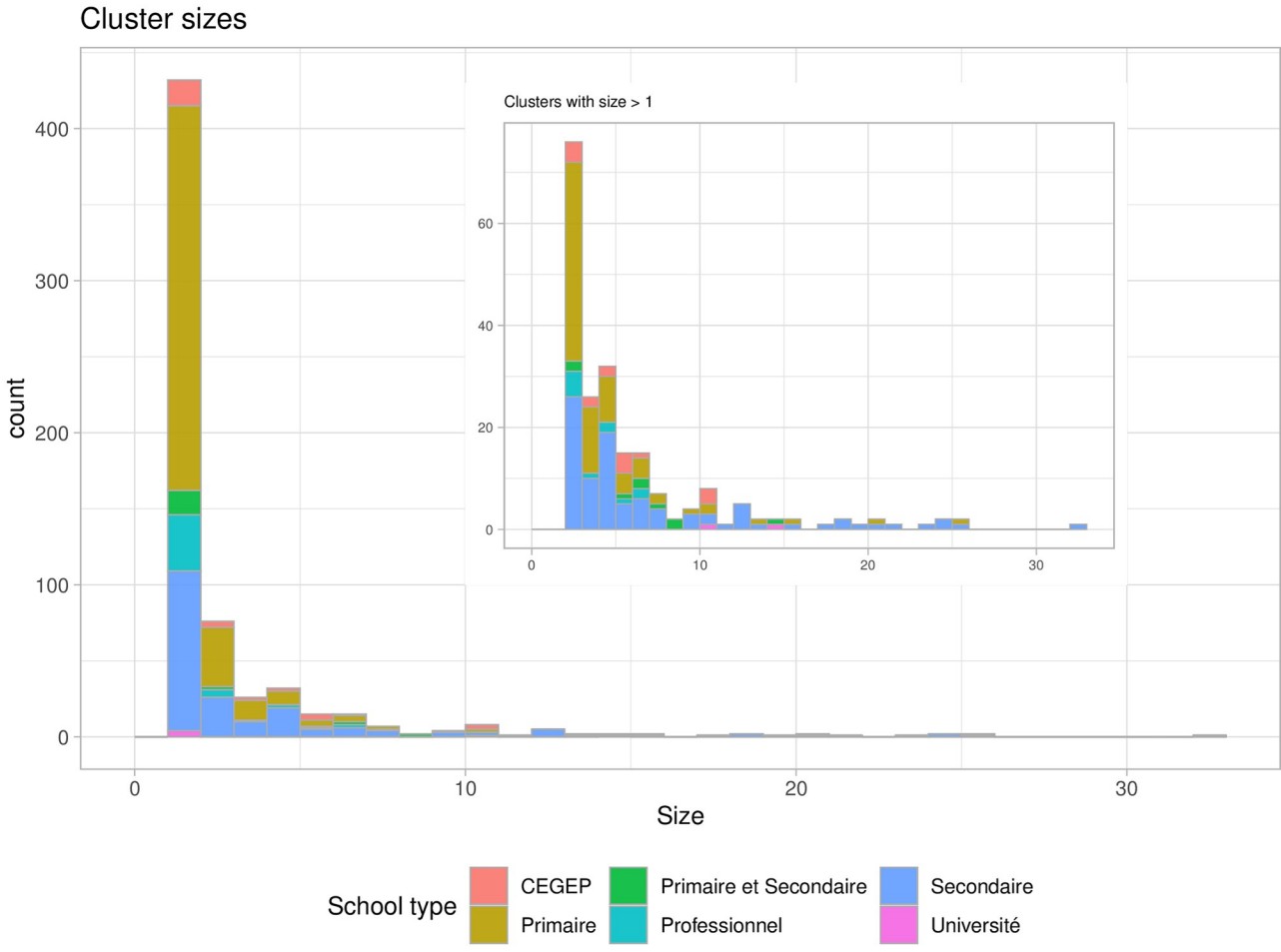

**Fig 1. Cluster sizes in Québec schools whose exposure was on or before Oct. 01, 2020, as of Oct. 11, 2020.** The inset shows only those with 2 or more cases; the main plot shows all exposures. Most exposures have not led to detected clusters; 33% of the exposures have led to at least one additional detected case, and 20% to at least two additional detected cases.

state, and ending when they recover. They enter the symptomatic state after the PIP, if they have not already recovered by that time. See the Supplemental Material for information about the parametrization and distribution of the latent period, the PIP, and the infectious period. With probability $\alpha$ they never show symptoms (and are asymptomatic); otherwise symptoms appear after the PIP.

Rather than assuming that all students in the class are equally likely to transmit the infection to each other, we model individual and contact group effects. We assume that the $n_{\text{class}}$ students are broken into smaller groups of $n_{\text{group}}$ students. We start with a base rate $\beta$ of transmission which represents the rate of transmission of from one infectious individual to another in the same group. Our default value for $\beta$ is 0.003 transmissions per contact per hour (0.006 in environments with increased transmission). Another source of variability is in the infectiousness of individuals. As discussed in the introduction, some evidence indicates that certain individuals are *superspreaders* and so have atypically large $\beta$ compared to others. The most important instance of this is when the index case has high $\beta$. In order to capture this, we model the index case as having a separate transmission rate $\beta_0 = f_{\text{index}} \beta$ where $f_{\text{index}} = 1$ or 3, depending on whether the index case has the same infectiousness or a higher infectiousness than

**Table 1. Model parameters.**

| Name | Symbol | value | units | citation |
|---|---|---|---|---|
| transmission rate | $\beta$ | 0.003–0.018 | per contact per hour | [38] |
| latent period mean, std | $\mu_\ell, \sigma_\ell$ | 3, 1 | days | [39] |
| PIP mean, std | $\mu_t, \sigma_t$ | 2, 0.5 | days | [40] |
| infectious period mean, std | $\mu_i \, \sigma_i$ | 10, 5 | days | [41] |
| probability asymptomatic | $\alpha$ | 0.4 | | [13, 22] |
| asymptomatic fraction of infectivity | $f_{\text{asymp}}$ | 0.8 | | |
| between-group fraction of infectivity | $f_{\text{aero}}$ | 0.25 | | |
| delay in testing | $t_{\text{delay}}$ | 2 | days | |

others. We also model reduced infectiousness of asymptomatic individuals. Their transmission rate is $f_{\text{asymp}} \beta$ where we choose $f_{\text{asymp}} = 0.8$. We explore the impact of these assumptions in the supplemental material. The final effect modifying transmission rate is to decrease it when the infectious person and the susceptible person are in different contact groups. The effect is to multiply $\beta$ by $f_{\text{aero}} = 0.25$. We model the effect of these different heterogeneities multiplicatively, so that if, for example, the index case is asymptomatic, the rate of transmission to a susceptible in another group is $f_{\text{index}} f_{\text{asymp}} f_{\text{aero}} \beta$. We note that our maximum value of $\beta$ (when both the environment and the infectiousness of the index case are most conducive to transmission) is 0.018 transmissions per contact per hour. This is considerably smaller than the estimates of index $\beta$ for widely-reported outbreaks in adults [38], by up to a factor of 30 for some events.

Table 1 lists the parameter values used in our simulations and provides supporting citations. We run our simulations twice: once with the index case is symptomatic, and once when the index case is asymptomatic, because this turns out to be a crucial factor in determining cluster size.

## Classroom structure

We model transmission in both an elementary school and a high school environment, taking the structures from that in British Columbia when schools opened in September 2020. For the elementary school, $n_{\text{class}} = 25$ students who spend 6 hours a day together, from Monday to Friday. All students except the index case are susceptible. The index case turns infectious at the beginning of the day on Monday. We assume that students are in 5 contact groups of $n_{\text{group}} = 5$. We simulate for 50 days, and in most simulations all students are recovered before the end of that period. For the high school, morning and afternoon are structured differently. In the morning $n_{\text{class}} = 30$ students in groups of size $n_{\text{group}} = 5$ meet for 2.5 hours Monday through Friday. In the afternoon $n_{\text{class}} = 15$ students meet in a distanced way for 2.5 hours on Tuesday and Friday only. We model the distancing using contact groups of size 1. We assume there is no overlap between the morning and afternoon classes except for the index case. Table 2 summarizes these classroom settings.

**Table 2. Classroom parameters.**

| Parameter | Elementary School | High School Morning | High School Afternoon |
|---|---|---|---|
| hours per day | 6 | 2.5 | 2.5 |
| number students | 25 | 30 | 15 |
| meeting days | Mon–Fri | Mon–Fri | Tue, Fri |
| students per group | 5 | 5 | 1 |

## Protocols

We consider four different protocols for what interventions are implemented when students become symptomatic or receive a positive test result. In each protocol students who become symptomatic immediately stop attending school and therefore cannot infect other students. (We do not model infection in the home environment, which is of course an important real-life practical consideration.) Every student who is symptomatic is tested and learns their results $t_{delay}$ = 2 days later. The value of the parameter $t_{delay}$ is important for the interventions, as the larger it is the more time pre- or asymptomatic students (who were infected by the index case, but remain in the class) have to infect their classmates. The protocols differ in which interventions are used after a student tests positive.

In the **baseline** protocol no further action is taken. Symptomatic students remain home and cannot infect other students, but the class continues to operate so that any other presymptomatic or asymptomatic students may infect others.

In the **contact** protocol as soon as a symptomatic student receives a positive test result all the other students in their group are isolated (sent home from the class) and no longer able to infect other students. It is possible for any number of groups to be isolated, and under this protocol those decisions are made independently.

In the **two groups is an outbreak** protocol, as in the contact model, groups with a student receiving a positive test are isolated, but when two or more groups are detected, an outbreak is declared and all students go into isolation, preventing any further transmission.

In the **whole class** protocol, when a symptomatic student receives a positive test result, all students are isolated and further transmission is prevented.

Fig 2 shows a cluster in an elementary school classroom and the effect of the four different protocols. Horizontal bars show the disease progression in students who are infected, with the index case at the bottom. Vertical black arrows show who infects whom, and the vertical grey bars indicate where different interventions take effect. In this particular simulation 20 out of 25 students are infected under the baseline protocol. Under the contact protocol this number is reduced to 12. Under the two groups is an outbreak protocol a single infection is prevented compared to the contact model by shutting the whole class earlier. The whole class protocol spares no additional infections in this case.

## Performance measures

For each of the protocols we consider three different performance metrics. *Total cluster size* is the number of students who are ultimately infected in class (or in both classes in the high school), including the index case. *Total disrupted* is the total number of students who are either asked to isolate or are tested. A student is included if they became symptomatic and had to isolate, if they were a member of a group that was asked to isolate, or when their class was asked to isolate (or be tested).

We did not explicitly simulate the number of new clusters that a cluster seeds through out-of-class social contacts (siblings, parents, teacher-teacher contact, after-school activities and so on). A measure of the risk of such "bridging" interactions is *asymptomatic student-days*, the total number of student-days when students are infectious, but are not isolating outside of the classroom. We assume that all symptomatic students are isolating outside the classroom, but whether asymptomatic students are depends on student behaviour, which is influenced by public health guidelines. We use the term *policy* to indicate the guidelines for student isolation outside the classroom, and its subsequent effect on student behaviour, (whereas we use *protocols* to refer to what happens in the classroom, as before.) Under a *lax* policy we assume that asymptomatic or presymptomatic students are not told to isolate (regardless of whether their

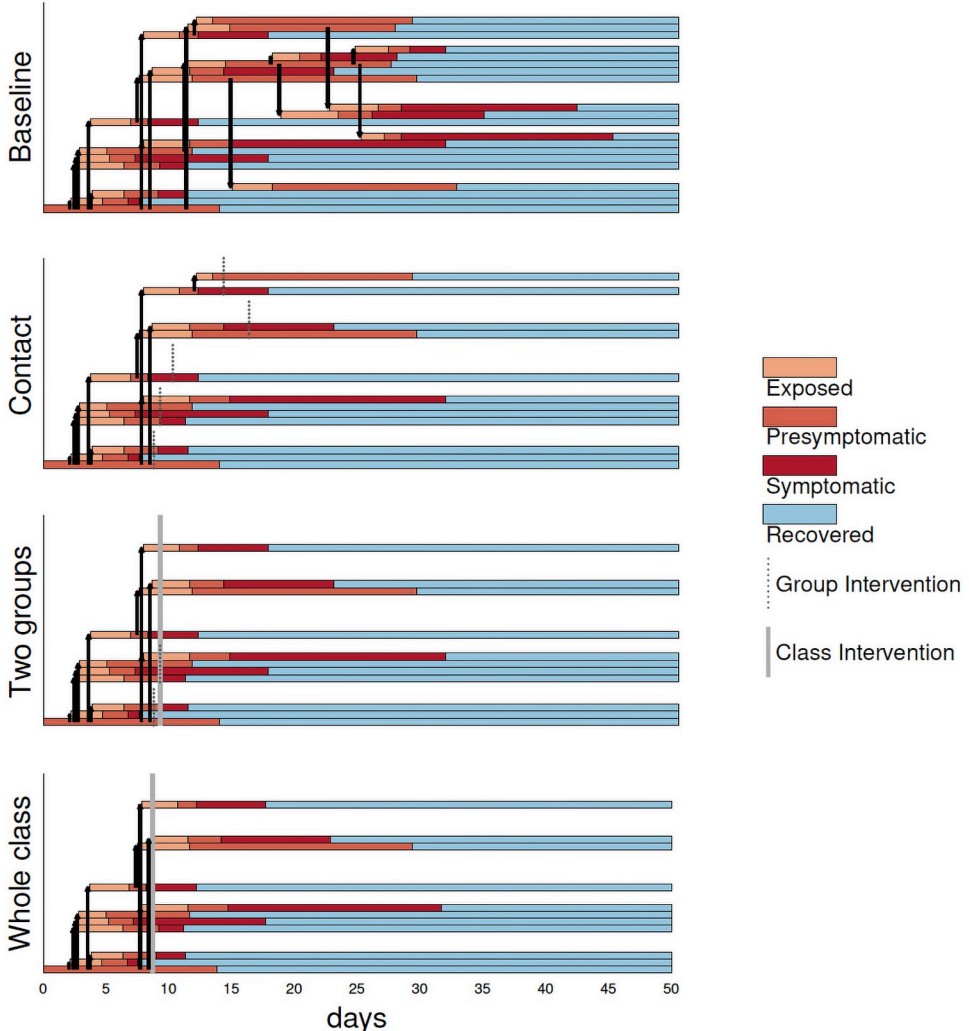

**Fig 2. The four protocols applied to a cluster where the index case is asymptomatic and has high transmission and where the class has medium transmission.** In all interventions, if individuals have not already been identified through the relevant protocol, transmission stops when symptoms begin (red to purple) as symptomatic individuals do not attend (or they leave when symptoms arise).

group or class is shut down), and the number of asymptomatic student-days is just the total number of student-days of infectiousness without symptoms. Under a *strict* policy we assume that when a group or class is shut down all students in the group are told to isolate until they recover or receive a negative test result. So the asymptomatic student-days under this policy is the total number of days students are infectious but asymptomatic *before* their group or class is shut down.

## Results

For four different combinations of class room $\beta$ and index case infectiousness we computed 1000 runs of the simulation for both a symptomatic and an asymptomatic index case. We show the distributions of our measures *total cluster size*, *total disrupted*, and *asymptomatic student-days* for a single introduced case in an elementary school in our model. See the supplementary material for the corresponding results for the high school model.

Fig 3 shows the distribution of total cluster size. The top left panels show results for when both the transmission in the class is low ($\beta$ = 0.003 transmissions per contact per hour) and the index case has the same transmission rate as others ($f_{index}$ = 1). Cluster size is small with a symptomatic index case (no transmission 73% of the time), but ranges from 1–5 individuals (median = 2) if the index is asymptomatic. This is because asymptomatic individuals have more time to expose others before recovering. None of the protocols make a large difference to the cluster size in this setting. If the index case has a higher infectiousness ($f_{index}$ = 3) but the room is still low risk ($\beta$ = 0.003 transmissions per contact per hour) (top right panels), again the cluster sizes are very small with a symptomatic index case (no transmission 53% of the time), though the tail of rare events is longer. When the index case is asymptomatic, in the baseline protocol the median cluster size is 5 and even in the whole class protocol, the median is reduced to 3. This pattern continues; with a highly infectious index case in a higher-risk room (bottom right): in the baseline protocol in which the main intervention is that symptomatic individuals do not attend, cluster sizes range from 0 to over 20 students in a single classroom (median = 4, sympt. index; median = 12, asymp. index). The whole class protocol reduces the mean cluster size from 11.9 to 6.5 in the aysmptomatic case, whereas the group and two group protocols reduce it to 8.3 and 7.5 students, respectively. Over all the scenarios the whole class protocol reduced cluster sizes roughly in half, with the contact and two group protocols doing a little worse.

Fig 4 shows the distribution of total disrupted for the four scenarios. The whole class model is the most disruptive, as expected. When transmission is low in the class and the index case is low risk, simply sending symptomatic individuals home accomplishes good cluster control and is least disruptive. In most transmission risk scenarios in which the index case is symptomatic, the median cluster sizes are small; however, there are rare high sizes in the long upper tail (for example, up to 20 students even with a low-risk classroom and medium-risk, symptomatic index case). These clusters can linger, eventually requiring each group to suffer disruption. In contrast, the whole-class model disrupts the whole class at the first positive test, leading to high levels of disruption and surprisingly weak control of larger clusters. This is particularly true when the index case is symptomatic.

Fig 5 shows the distribution of asymptomatic student-days in our four scenarios. By this measure, the effectiveness of the whole-class intervention is strong, particularly in the most unfortunate scenarios (high transmission index case and environment, and asymptomatic index). Here, the median numbers of asymptomatic student-days are reduced from 14.7 to 9.2 in the lax case and 2.5 in the strict case. In the low-transmission scenarios the whole-class intervention does remove the long tail (up to 50 student-days of potential infectiousness in the other protocols, compared to a maximum of less than 10 student-days in the whole-class model). Particularly in the "strict" case, the whole-class protocol achieves a dramatic reduction in the force of infection that can arise from a cluster, both in the median and in the variability, compared to the baseline protocol in which only symptomatic individuals cease attending. We note, however, that the two-group protocol (in which there is a whole-class-level intervention once two different contact groups have detected COVID-19 cases) achieves nearly the same level of reduction of the potential force of infection from the cluster, with less overall disruption.

We obtain qualitatively similar results for high schools, in which the model is more complex—see Figs C–F in S1 Text. However, because of the reduced duration of attendance in the high school configurations we modeled, the cluster sizes are smaller and less variable than they are in the elementary school model. The extent of disruption is higher due to larger numbers of overall contacts. Importantly the much lower cluster sizes in the high school setting versus the elementary point setting is due to how extensively the high school schedule has been

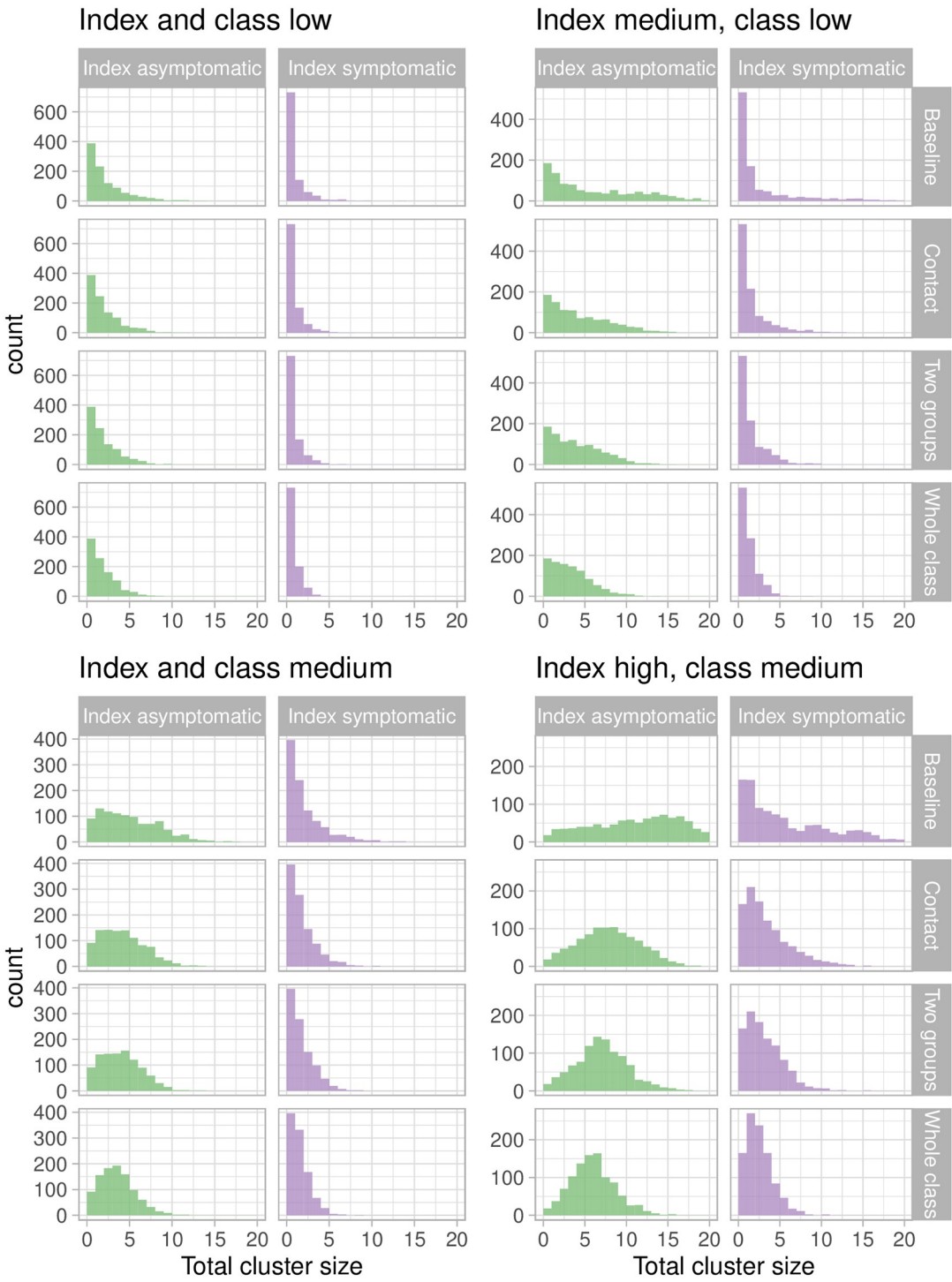

**Fig 3. Otal cluster size.** High variability in cluster sizes results from moderate variability in transmission. Cluster size distributions in 8 scenarios ranging from a low infectiousness index case in a low-transmission environment/activity (or class, top left) to an index case with 3 times the baseline transmission rate in an class with twice the baseline rate (bottom right). Left: the index case is asymptomatic. Right: the index case is symptomatic.

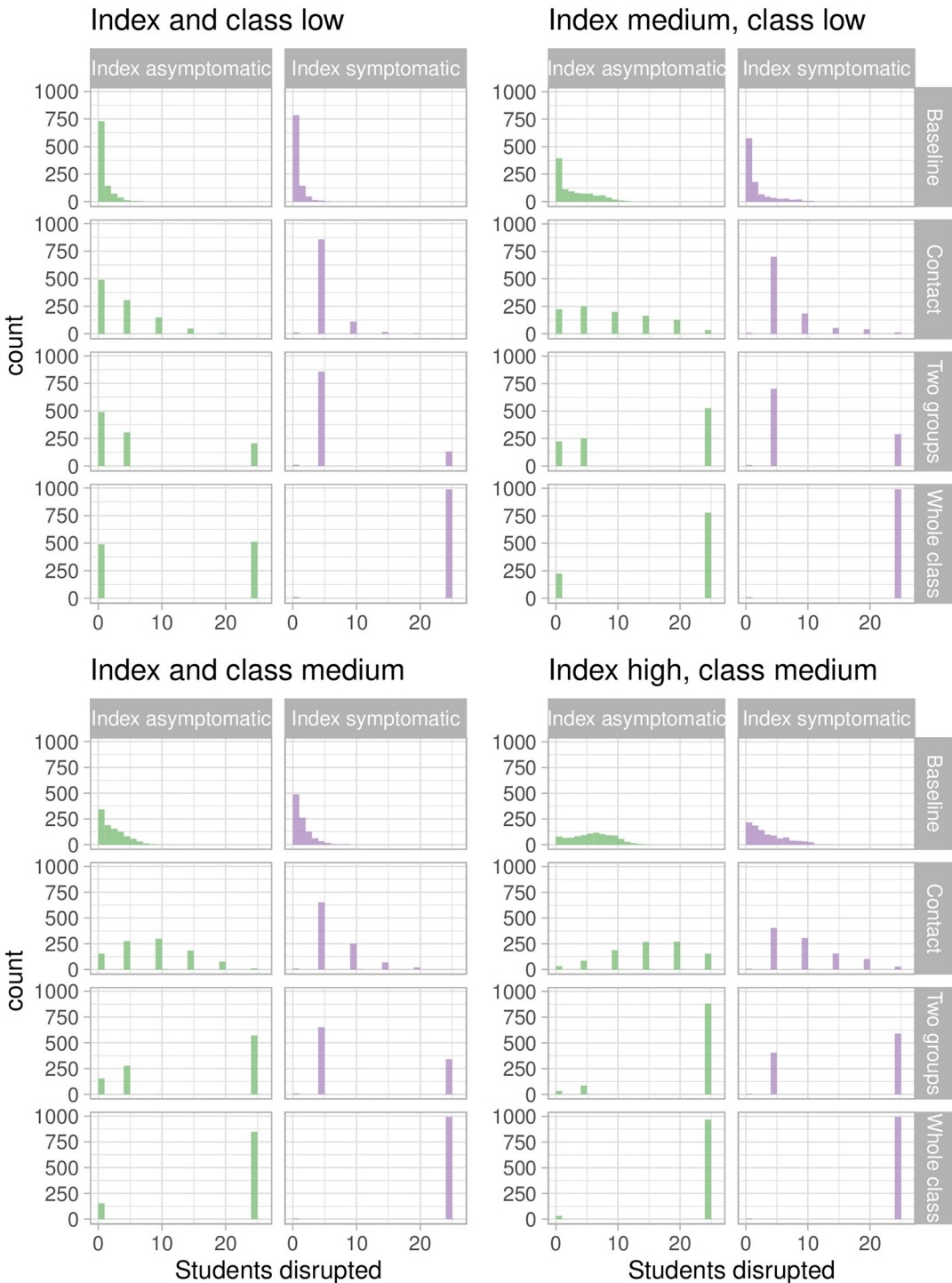

**Fig 4. Total disrupted: The total numbers of students who are either asked to isolate or must be tested, in the different protocols, according to the index and classroom's transmission risk and whether the index case is asymptomatic.** A student is included if they became symptomatic and had to isolate, if they were a member of a group that was asked to isolate or when their class was asked to isolate (or be tested).

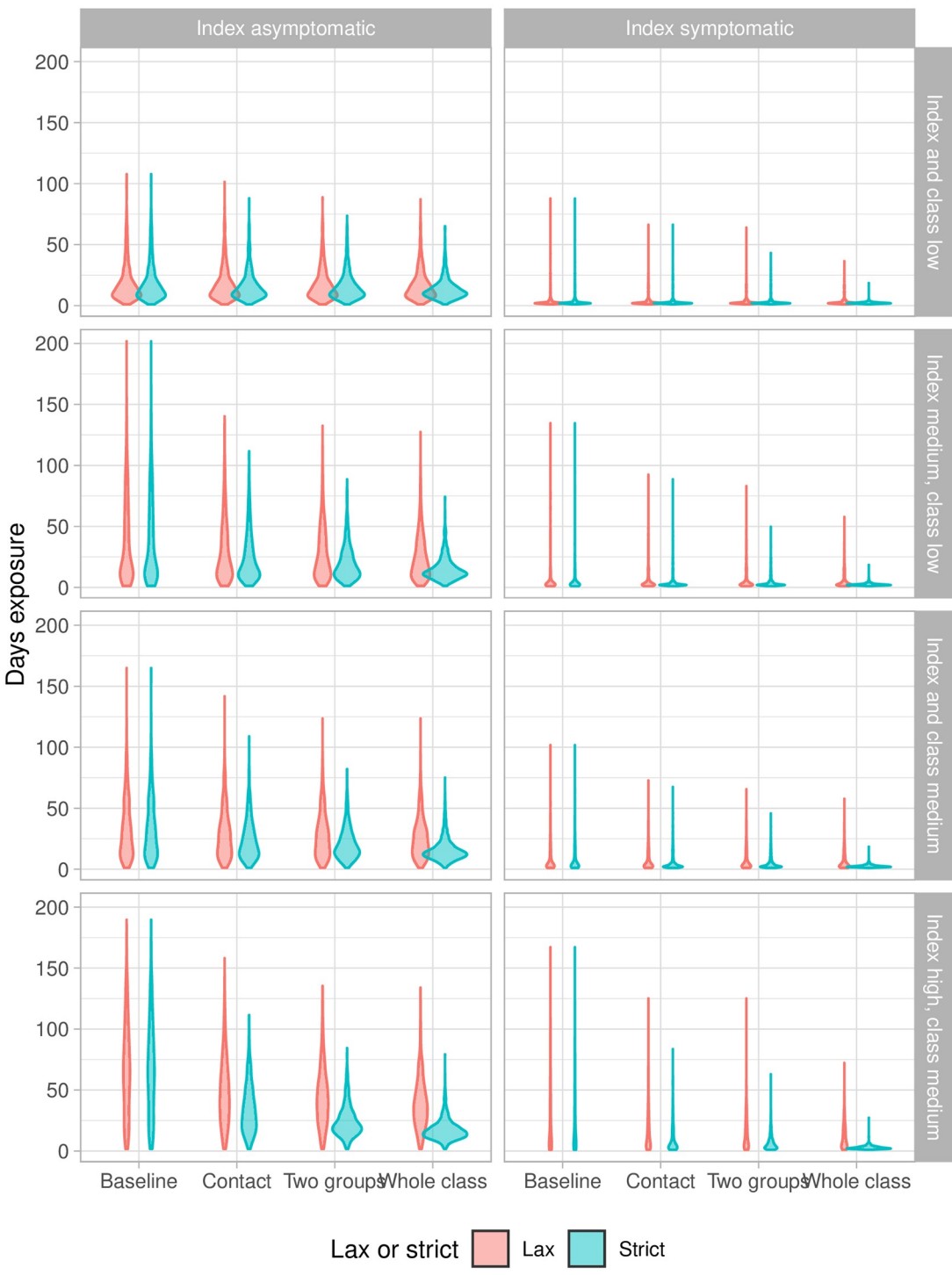

**Fig 5. Asymptomatic student-days: The number of student-days on which a student is infectious but not yet told to isolate.**
Left: index case is asymptomatic. Right: index case is symptomatic. Lax: Only symptomatic students are ever told to isolate.
Strict: All student in a shut down group or class are told to isolate.

restructured in response to the pandemic. We illustrate this by showing results on cluster size for a high school with pre-COVID structure: four 1.25 hour classes every day with largely different students in each. See Fig F in S1 Text.

The mitigation protocols in Fig 3 make a disappointing impact on the total cluster size; variation is driven much more by the transmission rate and whether the index case is asymptomatic. While the whole-class model in which testing (even asymptomatic) class members is used to identify infections rapidly reduces the number of student-days when infections go undetected, none of the protocols reduces the cluster sizes so greatly that they would be a reliable approach for in-class clusters in the unfortunate event where a highly infectious index arrives in a moderate-risk room. Fundamentally, this is because too much transmission can occur in the pre-infectious period if the index is symptomatic, and/or too much occurs before the first case has symptoms (if the index is asymptomatic). In most of our scenarios the index case directly infects most of the students who become infected, and so the amount of time the index case spends in class is key. If they are symptomatic, then the period of time they have to infect others is just the pre-infectious period, with an average of 1 or 2 days. But if they are asymptomatic they have the entire time until someone they infect becomes symptomatic to infect others.

Without closing schools down entirely, if we want to prevent large clusters from occurring altogether, this leaves approaches to detect potential index cases before they show symptoms. Pooled testing, wastewater monitoring and airflow monitoring have all been proposed with this aim [42, 43]. We simulated introduced cases and resulting transmission under the baseline of no regular testing (with the same baseline as above, symptomatic individuals going home) and compared this to weekly or every three day testing or environmental monitoring covering all individuals in the class. The results for the total cluster size are shown in Fig 6. Regular pooled or otherwise universal testing dramatically reduces the sizes of even the most unfortunate clusters (infectious index, higher-risk room), for example from a median of 12 to a median size of 3 if the index is asymptomatic. But even with regular pooled or otherwise universal testing, testing in a matter of hours (e.g. onsite) has a substantially greater impact than testing at a centralized laboratory (if that takes 2 days including shipping time).

## Discussion

Data on COVID-19 transmission in schools is consistent with overdispersed transmission in which many exposures—even a large majority—do not lead to clusters or outbreaks, but some do. Overdispersion in transmission is known in respiratory infectious disease and in COVID-19 in particular [30, 36, 44–46]. SARS-CoV-2 viral loads are reported to vary by over 11 orders of magnitude, with meta-analysis not finding significant differences in variability or viral load between children and adults [36], and with variation over the infectious period. Activities and symptoms both affect droplet production, and ventilation and distancing affect whether droplets or aerosols containing virions are likely to reach an individual. Our model structure captures this complexity using two components contributing to the transmission rate: host variability and a contribution from the environment and activity. In this framework, even relatively low variation in transmissibility between individuals, combined with even lower variation in the environment/activity's contribution, explains widely variable cluster sizes.

Our study has some limitations. We have not extended our simulations beyond the classroom (or high school classrooms) to simulate how each cluster may spread outwards via siblings, parents, teachers and their contacts, other household interactions, friendship groups and the broader community. These factors are complex and other models have explored them [47–51], some also finding that extensive testing or successful test and trace systems are required to

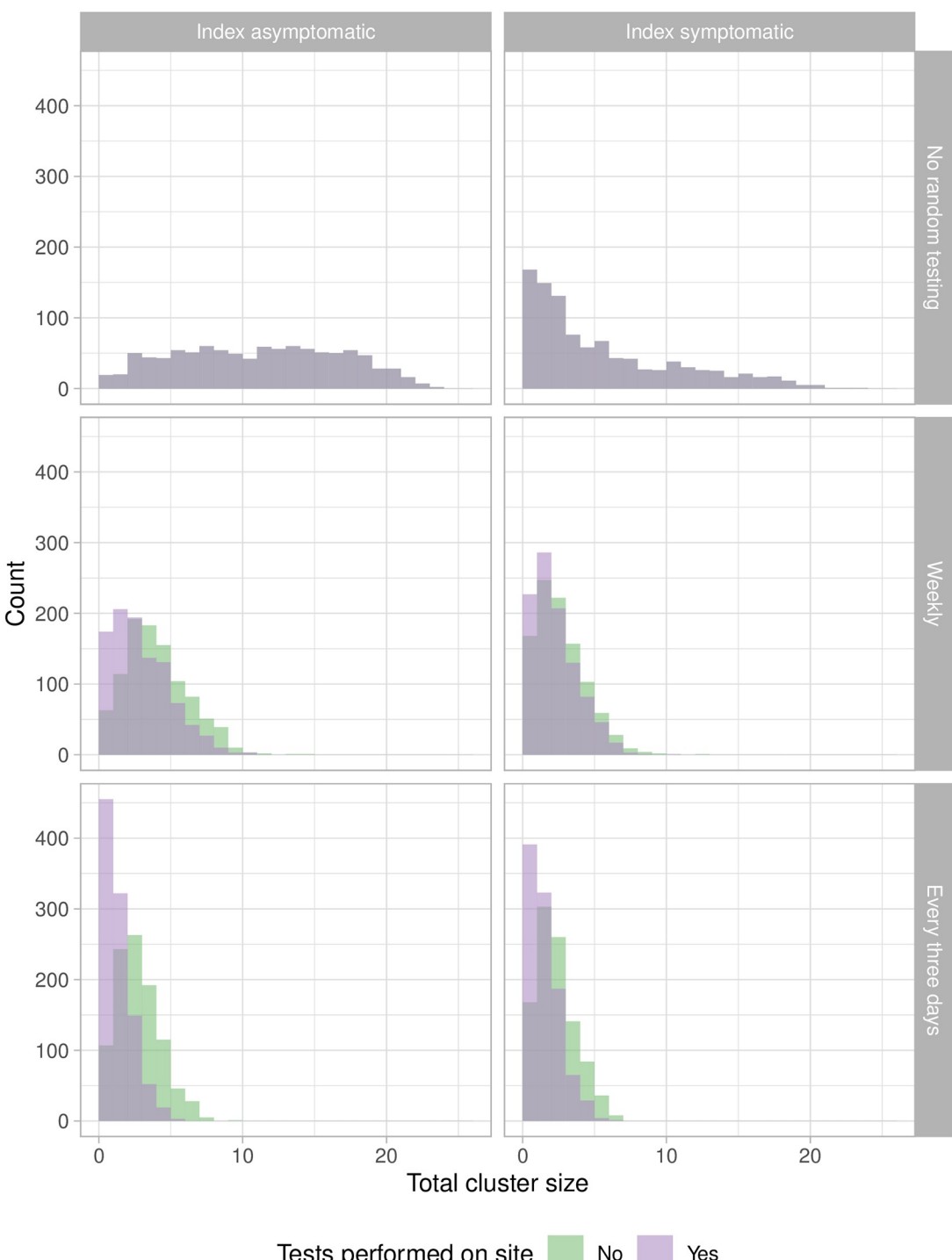

**Fig 6. Cluster sizes are greatly reduced with regular universal (e.g. pooled) testing, particularly when that testing is performed on site (in the model, in 2 hours, compared to an assumed 2-day time to result for tests processed off site).** Left: index case symptomatic. Right: index case asymptomatic. The baseline scenario shows the cluster sizes without regular testing, compared to weekly (middle row) or every 3 days (bottom row). Regular testing reduces the median cluster size from 4 or 12 (index symptomatic, asymptomatic) to 3 if testing is done offsite, or 2 if it is performed rapidly on site. The fraction of clusters of size > 5 is reduced from 80% to 20% (or 12% for rapid onsite testing) if the index is asymptomatic, and from 48% to just 2–3% if the index is symptomatic.

avoid schools amplifying COVID-19 transmission. We focused instead on how heterogeneity in transmission, arising from individual and environment effects, impacts the ability of mitigation measures to detect and control in-class transmission. We have a simple model of contact in which a known, fixed group of contacts are at highest risk from a given index case. This does not reflect the complex interactions in a classroom setting, but additional mixing or errors in identifying precisely who an index case was in close contact with can be modeled in the same way that we have modeled increased contribution of aerosol transmission (i.e. a risk of transmission outside the identified group of close contacts). There remain many unknowns about the timing and nature of COVID-19 transmission and we have used a simple model with constant infectiousness over time and with variability in the pre-infectious, infectious and symptomatic periods consistent with current knowledge of COVID-19 transmission.

In the particular context of schools, we find that interventions triggered by positive tests from symptomatic individuals are relatively ineffective in mitigating "unfortunate events"— high-transmission index cases in moderate (or higher)-transmission environments– even if everyone in the class is isolated upon the first positive test. There is growing evidence that large clusters can happen in schools and that children can transmit COVID-19 [30, 52]. This calls for preventive measures beyond protocols centred on symptomatic testing. Up to autumn 2020, school closures were the primary mechanism for preventing school transmission, and more broadly, widespread social distancing and non pharmaceutical interventions were the widely used and widely effective in controlling community transmission throughout spring 2020 [53–55]. If, instead, we are to maintain open schools, it is necessary to prevent large school transmission clusters, even if they are expected to be rare. The expected benefit of preventing large transmission clusters will naturally depend on the state of COVID-19 transmission in the community, with larger clusters likely to be amplified and spread onwards where community transmission is ongoing. Such settings will also have more school exposures, and the chance of an unfortunate high-transmission introduction to a school is correspondingly higher, creating a viscious cycle.

Our analysis and modeling suggest three approaches to prevention. First, reducing community transmission can play a large role; if exposures themselves are rare, the waiting time before a high-transmission introduction is likely to be much longer than if community transmission leads to frequent exposures. In a jurisdiction with 0.5% prevalence, where 75% of cases are symptomatic and not attending (or have been alerted to their exposure), the probability that a high school with 1500 staff and students has at least one case attending is still 85%. The more introductions happen in schools, the sooner we can expect to be unlucky. This may account for reports of large school clusters in Israel, Sweden, Chile [52] and some larger clusters in Québec [37], while countries with low overall levels found very low risk of transmission from children in the same period [4]. Second, testing can be used not only to mitigate one cluster in (e.g.) a classroom, but to prevent the next. We comment on two testing frameworks: testing triggered by detection of a symptomatic individual, and regular testing or monitoring to detect any COVID-19 in any individual, regardless of symptoms or known exposure (e.g. pooled testing). Rapid regular universal monitoring is far superior in preventing large clusters to testing that is initiated upon detection of a symptomatic case, even if a whole class is then tested soon afterward.

Finally, steps should be taken to control the environment's contribution to the variation in transmission rates (and therefore to cluster sizes). Indoor, crowded, loud, poorly ventilated environments with singing, eating and dining are recognized to be comparatively high risk [36, 38]. However, data could now be gathered prospectively with a focus on schools: when there are exposures in classroom settings, these could be linked to data on the room size, ventilation, whether windows were open, numbers of students in the class and classroom

organization, and then further linked to follow-up on cluster size. Less is known about what may lead an individual to have a high viral load and to generate high volumes of infectious droplets or aerosols, though symptomaticity (especially coughing) creates more droplets while talking, singing and breathing produce aerosols whether an individual is symptomatic or not [36]. Here too, data collection linking individual-level information with transmission via contact tracing and follow-up could aid in identifying risk and preventing high-risk introductions.

Our results have focused on classroom settings in schools but could apply to other settings in which people spend multiple hours per day with the same group of approximately 20–30 others, and have closest contact with a subset of these individuals. The fact that our results for the case of BC high schools (one in-person class per day with a hybrid class some afternoons) are very similar to the simpler contact structure in elementary schools indicates that the specific details of the contact patterns are less important for the cluster sizes and roles of mitigation than the variation in transmission. Accordingly, many workplaces may be well represented by our model and conclusions.

The recommendations we make here based on our simulation results have already played an important part in debates on how best to manage COVID-19 in schools and other similar environments [56–59]. Since our code is publicly available and well-documented, it can be modified rapidly to adapt it to other settings, as we recently did in order to model COVID-19 outbreaks in long-term care [60, 61].

## Supporting information

**S1 Text. In the document supplemental material we provide more details about several aspects of our work, including details of the simulation method, simulations with parameters set to capture high schools, and a set of results for the total cluster size with different choices for the mean pre-symptomatic infectious period, relative infectiousness of asymptomatic individuals, and extent of out-of-group mixing or aerosols.** Within S1 Text we have: **Table A**. Parameters for the three gamma-distributed time intervals in our model. **Fig A**. Distribution of latent period, PIP, and infectious period for our simulations. **Fig B**. Comparison of model results with analytical results in simplified setting. **Fig C**. Cluster sizes in a high school. **Fig D**. Students disrupted in a high school. **Fig E**. Student-days of undetected infection in a high school. **Fig F**. Comparison of cluster size in a pre-COVID high school with modified plan. **Fig G**. Cluster size under different parameter settings. **Fig H**. Cluster size under further different parameter settings.
(PDF)

## Acknowledgments

We thank Covid Écoles Québec for the data on cluster sizes in Québec schools.

## Author Contributions

**Conceptualization:** Paul Tupper, Caroline Colijn.

**Formal analysis:** Paul Tupper, Caroline Colijn.

**Methodology:** Paul Tupper, Caroline Colijn.

**Software:** Paul Tupper, Caroline Colijn.

**Visualization:** Paul Tupper, Caroline Colijn.

**Writing – original draft:** Paul Tupper, Caroline Colijn.

**Writing – review & editing:** Paul Tupper, Caroline Colijn.

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
