## [Decision Letter · Decision Letter 0]

3 Mar 2021

Dear Dr. Tupper,

Thank you very much for submitting your manuscript "COVID-19's unfortunate events in schools: mitigating classroom clusters in the context of variable transmission" for consideration at PLOS Computational Biology.

As with all papers reviewed by the journal, your manuscript was reviewed by members of the editorial board and by several independent reviewers. In light of the reviews (below this email), we would like to invite the resubmission of a significantly-revised version that takes into account the reviewers' comments.

We cannot make any decision about publication until we have seen the revised manuscript and your response to the reviewers' comments. Your revised manuscript is also likely to be sent to reviewers for further evaluation.

Sincerely,

Philip K Maini

Associate Editor

PLOS Computational Biology

Virginia Pitzer

Deputy Editor-in-Chief

PLOS Computational Biology

Reviewer's Responses to Questions

**Comments to the Authors:**

Reviewer #1: This paper looks at modelling COVID infections and possible interventions in the context of schools and the upset to education.

I have mixed feelings about this paper, but with a little work, I think I could be made publishable. Firstly, is this journal the right place for this paper? Being a computational journal I would have expected more discussion regarding the stochastic simulation. Further, it wouldn’t be too difficult to wrap some theory around these simulations either from a stochastic point, using a chemical master equations approach and Stochastic Simulation Algorithm, or as a system of ODEs. I understand the authors want to consider the stochastic effects, but the mean-field averages over 1000 simulations should start to provide results that could be compared with an ODE approximation. Essentially, providing these comparisons would give me more confidence in knowing that a large stochastic matlab simulation is doing the right thing.

But if the simulation is not the novelty, then perhaps the results are. However, we reach a sticking point again. Namely, the three results are: reduce contacts, provide better ventilation and, finally, testing is a great help. However, none of the insights are new, thus the authors should provide context for why their result is novel.

Overall, I hate to be negative as the paper is well written and I fully believe their results. Moreover, their introduction is an excellent review of current literature. However, at this time I feel that the article would be adding to the noise and rush of COVID publications, rather than boosting a signal that either provides a measurable impact, or a new idea that could tested. If the authors could add further information regarding their mathematical rigour as to make the simulations feel less ad hoc and/or provide context for why their results are needed, then I would be happy to change my mind.

Minor comments

Figure text is often too small to read.

In many figures the “y” axis is not labelled and so it is hard to understand what we are reading and whether the distributions are comparable.

The classroom parameters appear to be very specific to a particular school set up. How dependent are the results on this set up?

There are a few typographical errors throughout, for example lines 218 and 230.

Reviewer #2: Review for PCOMPBIOL-D-20-01998

COVID-19's unfortunate events in schools: mitigating classroom clusters in the context of variable transmission

Date: 15/12/20

Overview

=========

This paper uses simulated models of SARS-CoV-2 spread in schools to test whether simple sources of variability in individual epidemiological characteristics can explain the wide variety of cluster sizes seen in outbreaks in schools.

The outputs from the model are compared to data from Canadian schools.

A qualititative match between the schools data and the model is found.

The model is explained clearly and is appropriate for testing their main scientific question.

Minor points

I have no major criticisms of the analysis or model provided. I do have some minor comments.

This is largely subjective but I do not like the use of the word 'unfortunate', in the title and elsewhere, to describe large clusters. The assumption must be that these large clusters potentially lead to deaths after transmission from children to older or more vulnerable people and I don't think the word unfortunate handles this in a sensitive way. Relatedly, the grammar in the title sounds very odd to me due to using the possesive "COVID-19's". Simply "Superspreading events of COVID-19 in schools: mitigating classroom clusters in the context of variable transmission" or "Large clusters of COVID-19 in schools: ..." avoids the word unfortunate and sounds grammatically less odd.

A bit more detail on the Covid Ecoles Quebec data would be useful. Are the cluster sizes based only on testing symptomatic students? If so the cluster sizes are presumably underestimates. Given the qualitive nature of the comparisons being made, I don't think this is a big problem but a clear description would be useful.

Some plots (histograms or density plots) or summary statistics (95% intervals), perhaps in the supplementary material, for the gamma distributions used for the latent period, PIP and infectious period would be useful. At first look, the Gamma(mu = 10, sigma = 5) for infectious period seemed very unlikely to me. It took me quite a long time to examine the distribution further because there is no further details in the paper and the default gamma distributions in R use scale and shape. My choice of software is my problem of course but as these assumptions largely drive the entire conclusion of the paper, making them easy to examine is a good thing.

In many of the figures the two columns of subplots are labelled no and yes. This would be much clearer if they were "Index symptomatic" and "Index assymptomatic" or just "symptomatic" and "assymptomatic".

Line 218: "Symptomatic student are remain home and cannot" should be something like "Symptomatic students remain home

and cannot"

I found the interplay between {"baseline", "contact", "two groups is an outbreak", "whole class"} and {"lax", "strict"} quite confusing.

For example: in contact, "all the other students in their group are isolated". "Under a lax

policy we assume that asymptomatic or presympomatic students are never told to isolate". I don't understand how these two things can occur at the same time or if they don't I don't understand the relationship between policies and protocols.

**Have all data underlying the figures and results presented in the manuscript been provided?**

Reviewer #1: Yes

Reviewer #2: Yes

PLOS authors have the option to publish the peer review history of their article (what does this mean?). If published, this will include your full peer review and any attached files.

Reviewer #1: **Yes: **Thomas Woolley

Reviewer #2: No
---

## [Decision Letter · Decision Letter 1]

19 Apr 2021

Dear Dr. Tupper,

Thank you very much for submitting your manuscript "COVID-19 in schools: mitigating classroom clusters in the context of variable transmission" for consideration at PLOS Computational Biology. As with all papers reviewed by the journal, your manuscript was reviewed by members of the editorial board and by several independent reviewers. The reviewers appreciated the attention to an important topic. Based on the reviews, we are likely to accept this manuscript for publication, providing that you modify the manuscript according to the review recommendations.

Reviewer 1 still has substantial concerns, primarily about validating the accuracy of the numerical simulations and the novelty of the manuscript. However, we believe the issues raised by the reviewer can be addressed with a few more sentences that really clarify the issues, as well as attention to the minor comments regarding the figures.

Sincerely,

Philip K Maini

Associate Editor

PLOS Computational Biology

Virginia Pitzer

Deputy Editor-in-Chief

PLOS Computational Biology

[LINK]

Reviewer 1 still has substantial concerns, primarily about validating the accuracy of the numerical simulations and the novelty of the manuscript. However, we believe the issues raised by the reviewer can be addressed with a few more sentences that really clarify the issues, as well as attention to the minor comments regarding the figures.

Reviewer's Responses to Questions

**Comments to the Authors:**

Reviewer #1: Unfortunately, I am not satisfied with the authors' responses to my original queries.

My first comment was in regard to the amount of trust I can place in a computational simulation. I understand that they are focused on the stochastic features of the simulations. However, the ODE formulation was not intended to provide "accurate" results only provide confidence that the numerics are correctly implemented and doing what they are expected to be doing. I am happy for these results to be put in supplementary information, however, the results don't appear to compare well.

Specifically, the authors consider a simplified set up, which they say should match an analytical approach. Firstly, the text is not clear enough to reproduce how the results were produced, secondly, the left and right figures of (supplemental) Fig 2. don't appear to quantitatively correspond in the low transmission cases, where it is suggested that they should.

Since the authors cannot provide a simplified case that can be analytically checked, I have less confidence in the full simulation.

The second comment that has been ignored is that the y axes have still not been labelled with scales. The authors offer the response that

"The ggplot2 stat binline plot thatpermits splitting the histograms according to the intervention does not also allowa numeric value to show the numbers, buty-axis values are comparable frompanel to panel, and the overall scale is determined by the number of simulations making up the histogram."

To which my response is simply don't use ggplot2 then. Without knowing the values of the data we cannot hope to be able to undertstand the authors results clearly. Moreover, the histogram can be normalised to produce a probability density, which is independent of the number of simulations.

Equally one of my minor comments was ignored in that the text on many figures is still too small (supplemental figures too). As a rule of thumb the figure text should be no smaller than the caption.

Finally, one of my comments asked for a comment of novelty and impact. The authors have supplied this within the reviewer rebuttal, but didn't add anything to the document? Specifically, I would like to see a section near the discussion which emphasises the impact that the work has had. This will encourage a reader to use the work and codes supplied by the authors.

Reviewer #2: The authors have responded adequately to my original critique.

**Have the authors made all data and (if applicable) computational code underlying the findings in their manuscript fully available?**

Reviewer #2: Yes

PLOS authors have the option to publish the peer review history of their article (what does this mean?). If published, this will include your full peer review and any attached files.

Reviewer #1: **Yes: **Thomas E. Woolley

Reviewer #2: **Yes: **Tim Lucas

**Have all data underlying the figures and results presented in the manuscript been provided?**

Reviewer #1: Yes

Figure Files:

Data Requirements:

Reproducibility:

References:

---

## [Editor Report · Decision Letter 2]

27 May 2021

Dear Dr. Tupper,

We are pleased to inform you that your manuscript 'COVID-19 in schools: mitigating classroom clusters in the context of variable transmission' has been provisionally accepted for publication in PLOS Computational Biology.

Best regards,

Philip K Maini

Associate Editor

PLOS Computational Biology

Virginia Pitzer

Deputy Editor-in-Chief

PLOS Computational Biology

---

## [Editor Report · Acceptance letter]

18 Jun 2021

PCOMPBIOL-D-20-01998R2 

COVID-19 in schools: mitigating classroom clusters in the context of variable transmission

Dear Dr Tupper,

I am pleased to inform you that your manuscript has been formally accepted for publication in PLOS Computational Biology. Your manuscript is now with our production department and you will be notified of the publication date in due course.

With kind regards,

Zsofi Zombor
